# COVID-19 Pandemic Impact on Delays in Diagnosis and Treatment for Cervical Cancer in Montreal, Canada

**DOI:** 10.3390/curroncol32030147

**Published:** 2025-03-03

**Authors:** Yohan Kerbage, Elise Hillmann, Jessica Ruel-Laliberté, Vanessa Samouelian

**Affiliations:** 1Service de Chirurgie Gynécologique, CHU Lille, University Lille, 1 Avenue Oscar Lambret, F-59000 Lille, France; yohan.kerbage@chru-lille.fr; 2Centre de Recherche du CHUM (CRCHUM), Montréal, QC H2X 0A9, Canada; 3Gynecologic Oncology Division, Centre Hospitalier de l’Université de Montréal (CHUM), Montréal, QC H2X 3E4, Canada; 4Gynecologic Oncology Division, Centre Hospitalier de l’Université de Sherbrooke (CHUS), Sherbrooke, QC J1H 5N4, Canada; jessica.ruel-laliberte@usherbrooke.ca; 5Département d’Obstétrique-Gynécologie, Université de Montréal, Montréal, QC H3T 1J4, Canada

**Keywords:** cervical cancer, COVID-19, treatments delay, pandemic, volume reduction

## Abstract

Introduction. The COVID-19 pandemic has been responsible for a major reorganization of healthcare systems, with less access for cancer screening. Few data exist on the impact of cervical cancer treatment during the pandemic. Methods. The purpose of this study was to compare the cervical cancer stage at diagnosis and the surgical and medical treatment delays before and during the COVID-19 pandemic. This is a retrospective cohort study of all cervical cancers diagnosed at any stages between 1 January 2018 and 28 February 2022 at the Centre Hospitalier de l’Université de Montréal. Stage at diagnosis, time to initial referral, time from diagnosis to treatment before and during the COVID-19 pandemic were compared. Results. A total of 244 cervical cancers were diagnosed during the study period. No differences were observed between the number of cases diagnosed before and after pandemic (*p* = 0.237). Most patients and disease characteristics did not differ between the study periods, but the patients were significantly younger (*p* = 0.007), with higher BMI (*p* = 0.024) in the pandemic period. The mean time between initial diagnosis and referral was longer during the pandemic by 13 days (*p* = 0.042). The mean time between diagnosis and MRI and diagnosis and PET CT was not longer during the pandemic (*p* = 0.481 and *p* = 0.384). There were no significant differences in the mean time from the initial referring to the first visit at the CHUM (*p* = 0.895) or in the mean time from diagnosis to treatment (0.668) and duration of treatment (*p* = 0.181) Conclusion. Minor delays were observed during the COVID-19 pandemic. Cervical cancer patients treated at the CHUM, a tertiary and quaternary Canadian public health center, were globally referred and treated similarly, as those who were treated before pandemic.

## 1. Introduction

Cervical cancer is currently the 4th most common neoplasia in women worldwide; although, in Canada, it is the 15th most common cancer among females [1,2]. Still, in 2019, 1350 new cases were diagnosed, and 410 deaths were reported [1]. The therapeutic management of these patients is adapted according to the stage of the disease. These data, although very recent, are pre-pandemic data.

The COVID-19 pandemic has been responsible for a major reorganization of healthcare systems to deal with the massive influx of COVID-19 patients, to the detriment of care for common pathologies. The crisis organization was particularly developed during the first wave, resulting in a virtual paralysis of current activities in favor of the care of COVID-19 patients. This period of inertia then evolved towards a new organization, such as telemedicine, undermined with each new wave of the epidemic. Oncological gynecology has certainly paid a heavy price, but the consequences will probably continue, especially for the management of cervical cancer [1]. Indeed, due to its particular epidemiology (increased incidence in precarious populations), but also due to its accessibility to screening, monitoring, and treatment of precancerous lesions, it seems to represent a clinical situation particularly exposed during a pandemic.

The data in the literature are numerous, concerning the reorganization of screening policies, but also of treatment [3,4,5]. However, there is limited literature on epidemiological data during the pandemic [6]. In addition, the socio-economic level of each country as well as the heterogeneity of the organization of healthcare systems make it difficult to extrapolate the results.

The Centre Hospitalier de l’Université de Montréal (CHUM) is one of the main gynecological oncology reference centers in Quebec. This is one of the few studies to measure the impact of the pandemic on cervical cancer patients’ management in a large referral center. This study will make it possible to better adapt care services in the event of a similar episode in the future.

## 2. Materials and Methods

We carried out a single-center retrospective study at the CHUM from February 2018 to February 2022. All primary cervical cancers diagnosed in patients over 18 years, regardless of stage and histology diagnosed during this period, were included. Recurrent diseases, patients who have not received all or part of their treatment following diagnosis at the CHUM, patients who have only received brachytherapy (referred by other centers for lack of anesthetic resources) to the CHUM were excluded. Time of diagnosis corresponded to the time when the histological sample of cervical cancer was obtained, either by biopsy or by surgery. Within our center, consultation requests are centralized by an assistant and then prioritized by the on-call gynecologist–oncologist each week. That doctor is responsible for either the admission to the hospitalization unit of outpatients at the hospital or the patients consulting in the emergency room.

We considered the start of the pandemic as 12 March 2020, to carry out the before-and-after study. This is the date when the first governmental decisions, in response to the pandemic, were implemented in Quebec, and consequently at the CHUM. Data were collected retrospectively, using the institution’s electronic medical records. Demographic parameters and disease characteristics were collected along with the care trajectory key dates. Delays were calculated between the dates, and the differences were tested for significancy using the *t*-test. Delays from referral to consultation with a gynecologic oncologist were calculated by subtracting the time from referral to initial consultation at the CHUM. Time to treatment was calculated from the date of diagnosis (date of first cervical sample confirming cervical cancer), while time to realization of imaging was calculated from the day of the request for the imaging, until the date of the imaging. We used mean data imputation method for the missing data regarding the delay time to realization of imaging (PET CT and MRI), because more than 15% of the dates were missing. Treatment duration was defined by the time between the first treatment delivered, all techniques included (surgery, radiotherapy, chemotherapy, or a combination of those), and the end of treatment for early, locally advanced and metastatic cervical cancers.

Quantitative variables were described by mean and standard deviation or by median and interquartile range. The normality of the distributions was verified graphically and by the Shapiro–Wilk test. Qualitative variables were described via frequency and percentage. When the numbers were sufficient, the qualitative variables were compared between the groups using chi-square tests. If these tests were not valid (theoretical numbers < 5), Fisher’s exact tests were used. When the numbers were sufficient, quantitative variables were compared using Student’s *t*-tests. In cases of non-normality of the data, non-parametric Wilcoxon tests were used. The significance level was set at 0.05. Analyses were performed using SPSS software version 25 (SAS Institute, Cary, NC, USA). The study protocol was approved by the CR CHUM Ethics Committee under the number 2022-10360.

## 3. Results

A total of 244 newly diagnosed patients were registered during the study period (Figure 1). A total of 140 patients were registered during the pre-pandemic period (25.8 months), and 104 patients during the COVID-19 pandemic (23.5 months). No statistically significant difference was observed between the number of cases diagnosed before and during pandemic (*p* = 0.237). Concerning the general characteristics of the population (Table 1), no statistically significant difference before and during the pandemic was identified, except a younger population (*p* = 0.027), with a higher BMI (0.024), present during the pandemic.Figure 1Pancol-study design.
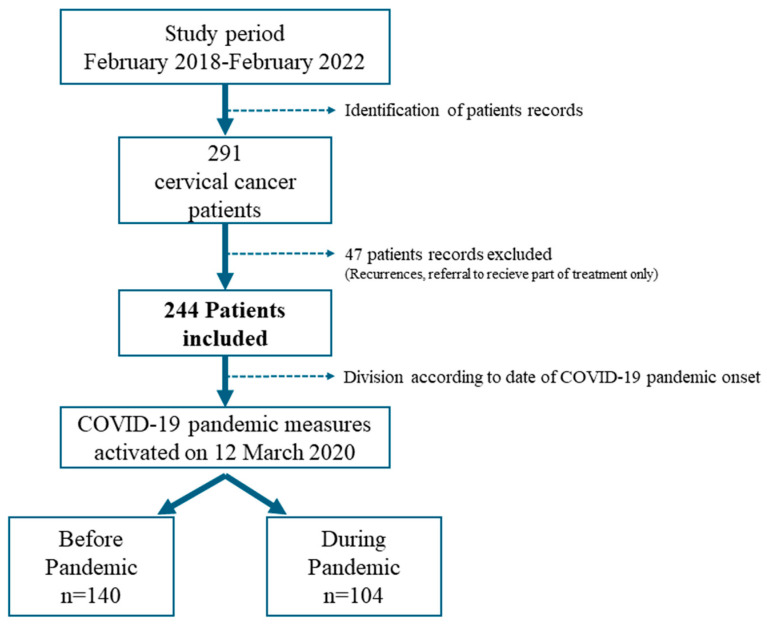


A higher proportion of patients during the pandemic had regular PAP smears (0.042). Stage II at diagnosis was more frequent before the pandemic, but no differences were observed regarding the diagnoses at all stages (*p* = 0.149). The proportion of Canadian born patients did not differ in the study periods (*p* = 0.124). Not surprisingly, in line with previous results, all modalities of treatment were comparable before and during the pandemic (Table 2). The surgical type differed between pre- and post-pandemic (*p* = 0.047), with less cone biopsies being performed during the pandemic. A higher number of laparotomies were also performed during the pandemic (*p* = 0.034).

For the patients who received only chemotherapy, significant differences were observed regarding the adjunction of bevazumab during the pandemic (*p* = 0.010). For patients who received chemotherapy concomitant to other treatment, significant differences were observed (*p* = 0.012); Cisplatin, with or without the adjunction of pembrolizumab, was higher during pandemic, and carboplatin was less adopted. For more advanced stages, differences regarding the systemic therapies were not observed.

The time between diagnosis and referral increased during the pandemic by an average of 13 days (*p* = 0.042). Time between the referral and the first consultation in our center remained stable, even with the exclusion of patients referred by non-oncologist practitioners working in our center (Table 3). Concerning imaging exams, time to PET scanner realization and time to MRI remained stable (*p* = 0.384 and *p* = 0.481). The mean delay time between the diagnosis and the beginning of treatment did not differ (*p* = 0.668). The mean duration of treatment was also not modified between the two periods (*p* = 0.181).

**Table 3 curroncol-32-00147-t003:** Timing of overall cases.

	Before Pandemic (%)	During Pandemic (%)	*p*
Overall numbers	140	104	0.237
Time between referral and first consultation at the CHUM (days)	21.9 ± 18.5	22.0 ± 41.5	0.998
Time between referral and first consultation at the CHUM, without patients referred by MD working at the CHUM (days)	23.7 ± 18.1	23.1 ± 42.3	0.895
Time between referral and diagnosis (days)	15.8 ± 32.6	28.2 ± 55.2	0.042
Time between request and PET scan (days)	13.3 ± 14.6	11.7 ± 10.9	0.384
Time between request and MRI (days)	16.9 ± 12.0	18.1 ± 13.6	0.481
Time between diagnosis and beginning of treatment (days)	88.7 ± 49.1	91.7 ± 57.8	0.668
Treatment duration (days)	37.0 ± 41.7	46.8 ± 65.1	0.181

## 4. Discussion

The impact of the pandemic on the healthcare system is undeniable, and has led to a restructuring of the healthcare system, with profound adjustments depending on the number of cases and hospitalizations. It was expected that this pandemic would have a negative impact on the management of cancers, especially uterine cervical cancer, due to a reduced access to healthcare facilities. However, our study, conducted in a single tertiary reference center, has shown a slightly comparable management before and during the pandemic. The main difference was related to the time between diagnosis and referral. Nevertheless, neither did the time between the diagnosis and the treatment onset, nor the treatment duration or the type of treatment, differ between the pre- and post-pandemic periods.

We did not observe a reduction in newly diagnosed cervical cancer during the pandemic. Our results are discordant with the two most recent studies, published in the Romanian and UK populations [7,8]. Popescu et al. published a study, in 2022, on the Romanian population. Unlike our study, they also looked at screening and diagnosis methods of cervical cancer during the pandemic. Over the pre-pandemic period, 2018–2020, compared to the pandemic period, 2020–2022, a 45% significant difference was observed in the number of cancers diagnosed at all stages [9]. In the study, published by Davies et al., the difference in the stage at diagnosis was also significant between the stages, with more stage II cancers diagnosed before the pandemic, and more stage III cancers diagnosed after the pandemic (*p* = 0.04) [7]. On the other hand, a Brazilian study conducted over a shorter period, from September 2019 to January 2020 and from September 2020 to January 2021, did not show any significant differences [6]. The difficulty in comparing these results lies in the difference in the organization of care between the different countries. Gynecologic oncology in Canada is extremely centralized, which may explain our results. Oncology services were prioritized during the pandemic in Quebec and across Canada, while the treatment of benign conditions was put on hold. The pandemic patients diagnosed with cervical cancer were more likely to have had regular PAP smears, although both proportions (pre-pandemic and pandemic) were extremely low (20.3% and 32.2%, respectively) for a developed, high-income country. This is resulting from the lack of a cervical screening program and from the challenges to access first-line healthcare and PAP smears. A novel approach to screening, using HPV-testing and allowing for self-collection, with a proven accuracy, should help improve the screening efficacy [10].

Regarding the time to first consultation in gynecologic oncology at our center, we did not notice any significant difference between the pre-pandemic and pandemic period. Frey et al. published a study finding an overall rate of modification of treatment for gynecological cancers of 38.7%, and a modification of 67.4% for those scheduled for surgery, of 21.5% for those scheduled for chemotherapy, and 18.8% for those scheduled for radiation [8]. This study included both new diagnoses and recurrences. There was no subgroup analysis by type of cancer, but the multivariate analysis showed no significant differences between new diagnoses and recurrences. Frey et al., similarly to Piedimonte et al., included several centers (both public and private in the Piedimonte study) and was not dedicated only to cervical cancer cares [8,11].

Regarding cervical cancer, Davies et al. reported only on the time between symptomatology and first consultation, for which no significant difference was demonstrated (*p* = 0.7) [7] We did not have access to these data in our study. Popescu et al. and Bonadio et al. reported a significant difference in pre-pandemic and pandemic time between the diagnosis and the first consultation in gynecologic oncology [6,9]. However, the delays, even pre-pandemic, seemed long: 4.1 [2,11] vs. 6.4 [3,12] and 4 months vs. 6.1 months, respectively. Literature is not definitive in evaluating whether a difference of 2 months is clinically significant at any stage, because there is no published analysis on the oncological outcomes according to the stage and the time needed to start treatments after consultation in gynecologic oncology.

Regarding the accessibility to imaging services, there are no publications on the subject, apart from screening imaging, particularly breast imaging. A report from the Canadian Radiology Resilience Taskforce made it possible to draw up a picture of the imaging procedures carried out during the pandemic, with a clear decline from 100% to 52% in CT scans, and from 100% to 40% in MRIs, between February 2020 and April 2020 [12]. This can be attributed to longer procedures (sterilization), but also probably to the deprogramming of the examinations deemed less urgent (follow-up, benign pathologies), the reduction in indications caused by fewer consultations, and the reduction in available personnel. On the other hand, oncologic patients had their access prioritized during the pandemic, due to Quebec’s public health policies. This is confirmed by our results concerning access to MRIs and PET CTs.

Some data from our study differ between the pre-pandemic and pandemic period. For early cancers, more cone biopsies were performed pre-pandemic. This result is explained by early cancer stage IAs that were treated differently based on age and fertility sparing. In addition, more patients have benefited from a laparotomy, which resulted from a change in the practices in our center, following the publication of the LACC trial [13]. For more advanced cancers, the addition of bevacizumab was recently introduced in our center and can explain the (non-significant) differences in its use.

Whether the differences in the various studies concerning the management of newly diagnosed cervical cancers are observed or not, the latent question is the survival of the patients diagnosed during the pandemic, and the financial and structural means engaged by public policy. The study by Popescu et al. showed a very significant decrease of almost 75% in April 2020, during the first lockdown, after which the volume of cases decreased by up to 36.1% in December 2021. The total volume loss of the tests conducted during the first 24 months of the pandemic was 49.9% [9]. A study taking place in the US shows a reduction of more than 70% in PAP smears [14]. Similar data are observed regarding HPV testing, with a reduction of 80% among women aged 30–65 years. Other studies focused on different cancer types that are dependent on screening programs to preserve good curative results, such as breast cancer and colorectal cancer. All observations converged to the same concerning fact, that the COVID-19 pandemic caused a steep decline in screening and diagnosis availability, with a negative forecast in long-term cancer outcomes [15,16,17]. There was also an important decrease in the number of patients included in clinical trials during the pandemic, which was not measured in our study, but which has been shown by Ali et al. in the United Kingdom [15].

The limitations of the present study are the retrospective nature of the study design, including the single-center design, and data availability from the patient records. Data collection from a single-center may not be representative for the entire Canadian population, as proportions of patient background characteristics may vary with important differences across provinces. The strengths of our study include the large volume of patients treated in our tertiary and quaternary gynecologic oncology cancer centers. The representativity of our sample is significant in regard to the Quebec’s female population.

## 5. Conclusions

Minor delays were observed during the COVID-19 pandemic. Cervical cancer patients treated at a tertiary and quaternary Canadian public center were globally referred and treated similarly to those who were treated before pandemic. Nonetheless, the real impact should be interpreted using data from specific countries and healthcare organizations. Screenings and vaccinations are the keystones of preventing cervical cancer, and the real impact of the pandemic will present itself in those two parameters within the next five years; however, it could be prevented by an appropriate political decision regarding the prevention and detection of early-stage cancer.

## Figures and Tables

**Table 1 curroncol-32-00147-t001:** Patients and disease characteristics.

	Before Pandemic (%)	During Pandemic (%)	*p*
Overall numbers	140	104	0.237
Canadian born			0.124
Yes	121 (88.3%)	77 (81.1%)	NS
No	16 (11.7%)	18 (18.9%)	NS
Age	51.8 ± 13.7	47.1 ± 12.9	0.007
BMI (kg/m^2^)	25.4 ± 6.1	27.2 ± 5.9	0.024
Regular PAP smears			0.042
No	110 (79.7%)	61 (67.8%)	S
Yes	28 (20.3%)	29 (32.2%)	S
FIGO Stage 2018			0.149
I	64 (45.7%)	59 (56.7%)	NS
II	19 (13.6%)	6 (5.8%)	S
III	34 (24.3%)	25 (24%)	NS
IV	23 (16.4%)	14 (13.5%)	NS
Treatments			0.373
Surgery	52 (37.1%)	48 (46.2%)	NS
Radio-chemotherapy + brachytherapy without surgery	56 (40.0%)	42 (40.4%)	NS
Radio-chemotherapy + brachytherapy with surgery	1 (0.7%)	0 (0.0%)	NS
Neoadjuvant chemotherapy	1 (0.7%)	0 (0.0%)	NS
Palliative chemotherapy	16 (11.4%)	10 (9.6%)	NS
Palliative radiotherapy +/− chemotherapy	5 (3.6%)	1 (1.0%)	NS
Palliative care only	4 (2.9%)	0 (0.0%)	NS
Others	5 (3.6%)	3 (2.9%)	NS
Quebec’s region			0.417
Abitibi-Témiscamingue	5 (3.6%)	5 (4.8%)	NS
Lanaudière	23 (16.4%)	10 (9.6%)	NS
Laurentides	12 (8.6%)	11 (10.6%)	NS
Laval	2 (1.4%)	4 (3.8%)	NS
Mauricie et Centre-du-Québec	1 (0.7%)	0 (0.0%)	NS
Montérégie	45 (32.1%)	44 (42.3%)	NS
Montréal-Centre	49 (35.0%)	29 (27.9%)	NS
Nord-du-Québec	1 (0.7%)	0 (0.0%)	NS
Outaouais	2 (1.4%)	1 (1.0%)	NS
Tobacco use			0.679
No	66 (47.1%)	50 (51.5%)	NS
Yes	54 (38.6%)	32 (33.0%)	NS
Ex-smoker	20 (14.3%)	15 (15.5%)	NS

**Table 2 curroncol-32-00147-t002:** Treatments details.

	Before Pandemic (%)	During Pandemic (%)	*p*
Overall numbers	140	104	0.237
Type of surgery			0.047
Radical hysterectomy	22 (41.5%)	22 (44.0%)	NS
Total hysterectomy	20 (37.7%)	21 (42.0%)	NS
Trachelectomy	1 (1.9%)	5 (10.0%)	NS
Cone biopsy	10 (18.9%)	2 (4.0%)	S
Adjuvant treatments			0.881
No	66 (93.0%)	87 (93.5%)	NS
Yes	5 (7.0%)	6 (6.5%)	NS
Type of adjuvant treatments			NS
Pelvic radiotherapy	4 (80.0%)	3 (50.0%)	NS
Radiotherapy and chemotherapy	0 (0.0%)	2 (33.3%)	NS
Radiotherapy and brachytherapy	1 (20.0%)	0 (0.0%)	NS
Chemotherapy only	0 (0.0%)	1 (16.7%)	NS
Chemotherapy associated with radiations			0.012
Cisplatin	51 (83.6%)	42 (80.8%)	NS
Carboplatin	8 (13.1)	1 (1.9%)	S
Cisplatin and carboplatin (intolerance or toxicity of cisplatin)	2 (3.3%)	5 (9.6%)	NS
Cisplatin +/− pembrolizumab	0 (0.0%)	4 (7.7%)	S
Neoadjuvant chemotherapy			0.175
Cisplatin placlitaxel	1 (16.7%)	0 (0.0%)	NS
Carboplatin placlitaxel	3 (50.0)	5 (62.5%)	NS
Cisplatin + paclitaxel + bevacizumab	0 (0.0%)	2 (25.0%)	NS
Carboplatin + etoposide	0 (0.0%)	1 (12.5%)	NS
Cisplatine + etoposide	2 (33.3%)	0 (0.0%)	NS
Chemotherapy only			0.010
Cisplatin + placlitaxel Carboplatin + placlitaxel	10 (52.6%)	1 (20.0%)	NS
Cisplatin + paclitaxel + bevacizumab	0 (0.0%)	3 (60.0%)	S
Carboplatin	1 (5.3%)	0 (0.0%)	NS
Carboplatin + paclitaxel +/− pembrolizumab	2 (10.5%)	0 (0.0%)	NS
Carboplatin+placlitaxel+bevacizumab	0 (0.0%)	1 (20.0%)	S
Cisplatin + paclitaxel +/− pembrolizumab	1 (5.3%)	0 (0.0%)	NS
Cisplatin+placlitaxel+ bevacizumab+pembrolizumab	3 (15.8%)	0 (0.0%)	NS

## Data Availability

The data that support the findings of this study are available on request from the corresponding author. The data are not publicly available due to privacy or ethical restrictions.

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
