# Peer review of "COVID-19 Pandemic Impact on Delays in Diagnosis and Treatment for Cervical Cancer in Montreal, Canada"

_curroncol, 2025, doi:10.3390/curroncol32030147_

Round 1

Reviewer 1 Report

Comments and Suggestions for Authors

This study investigated the impact on delays in diagnosis and treatment for cervical cancer in Montreal, Canada. Although the COVID-19 pandemic is currently coming to an end, and diagnosis and treatment is returning to normal, a new infectious disease pandemic may occur in the future. Therefore, it is important to investigate how the unique conditions of this COVID-19 pandemic have affected the diagnosis and treatment of a potentially critical but less urgent disease: cancer.  

While I believe this study is important, I would like to mention a few areas where I believe there is room for improvement.

1.      In the Materials and Methods section, it is stated that the mean data imputation was used for the missing data regarding the delay’s time to realization of imaging, but I am concerned about outliers. Although Table 3 shows some ranges, I would like to see actual data information including maximum and minimum values, for example in a histogram.

2.      Did the ±values in Table 3 indicate SD? Please consider making the tables self-contained by adding footnotes.

3.      Since the proportions of stages or treatment modalities were not perfectly the same, it didn’t seem reasonable to compare treatment duration before and after the pandemic for the cohort including all cases. Is it possible to compare surgery vs. surgery, or radiation vs. radiation? Was there any change in radiation therapy during the pandemic, such as a preference for shorter regimens?

4.      The study period is 25.8 months before the pandemic and 23.5 months after the pandemic. Since you are comparing the number of patients before and after the pandemic, I think the periods should be the same.

5.      There was a significant difference in BMI and age before and after the COVID-19 pandemic. I think it would be better if you mentioned the reason for this in the section of Discussion.

Author Response

  1. In the Materials and Methods section, it is stated that the mean data imputation was used for the missing data regarding the delay’s time to realization of imaging, but I am concerned about outliers. Although Table 3 shows some ranges, I would like to see actual data information including maximum and minimum values, for example in a histogram.
  • We only imputed value where we had the patient image exam but no access to the examen request date. Patients which underwent treatment without PET scan or MRI did not have data imputed, they were censored. So, for the PET scan we had 18 dates missing and for MRI we had 36 missing dates. We calculated t-test for both scenarios and found no difference between imputed and censored data calculations. Table and column graphics are presented on excel file Tab “Data imputation vs data censoring”. 
  1. Did the ±values in Table 3 indicate SD? Please consider making the tables self-contained by adding footnotes. 
  • Thank you, all column headers and footnotes were revised and added.
  1. Since the proportions of stages or treatment modalities were not perfectly the same, it didn’t seem reasonable to compare treatment duration before and after the pandemic for the cohort including all cases. Is it possible to compare surgery vs. surgery, or radiation vs. radiation? Was there any change in radiation therapy during the pandemic, such as a preference for shorter regimens?
  • The comparison of the proportions of stages and treatments are presented in Table 1. For treatments p is 0.373 and all categories presented no significant difference between before and during pandemic, while for disease stages p was non-significant (p= 0.149) and only the proportion of stage II patients were different while all other categories were similar. Therefore, we proceed with the analysis, including all cases. 
  • We opted for not further stratifying the cohort since the sample size was not optimized for all categories, these would lead to a loss of analysis power. 
  • No protocols for shorter radiation therapy regimens were put in place, since oncologic treatment was prioritized and the were no need to further reduce access to oncologic patients during the pandemic.
  1. The study period is 25.8 months before the pandemic and 23.5 months after the pandemic. Since you are comparing the number of patients before and after the pandemic, I think the periods should be the same.
  • Thank you for your comment. For better equity of the study periods, you are right, unfortunately the study period choice was not based on duration before and after pandemic onset, but on calendar Months. The study included patients from February 2018 to February 2022.
  1. There was a significant difference in BMI and age before and after the COVID-19 pandemic. I think it would be better if you mentioned the reason for this in the section of Discussion.
  • You are right to mention that there is a statistical difference between BMI and age, before and during Pandemic. Nevertheless, this difference is not clinically significant to suggest a different clinical approach to treat those patients. A phrase on the results section was included. 

Reviewer 2 Report

Comments and Suggestions for Authors

This is an interesting article that provides local data on the impact of the COVID-19 pandemic on cervical cancer treatment. I have some suggestions that may improve this article.

Materials and Methods: lines 65-68: Do you know exactly how many patients were excluded because they had only received brachytherapy at CHUM or had only part of their treatment there? It may be helpful to have a figure that displays how many patients were excluded due to various reasons.

lines 80-81: "Date of referral to gynecologic oncology at our center, and date of first consultation by a gynecologic oncologist"; For readability, it may make sense to combine this statement with the previous sentence or edit it to be a complete sentence. 

lines 82-84: Regarding the date of diagnosis, can you clarify the source for this information? Is it obtained from the referring institution if a patient is referred to CHUM for treatment or further evaluation? 

line 98: The software reference is for SAS, but I believe IBM still owns SPSS. Please clarify.

Lines 134-135: This sentence seems to be a repeat of the sentence at the beginning of the paragraph (at line 130). 

Table 2: The formatting in this table needs to be fixed so that the data align with the appropriate category labels. 

Discussion: First paragraph: Given this is a single institution study, you may want to frame this paragraph in reference to your experience at a tertiary institution. Additionally, are you able to provide any data that shows your institution is treating most women diagnosed with cervical cancer in a defined geographic area? That may help with generalizability of findings within your geographic region.

lines 221-223: "There was also an important decrease in the number of patients inclusion in the protocol..." I was confused by what you mean here. Is there a way to restate this sentence? "patients included in the protocol?" 

Comments on the Quality of English Language

"

Author Response

This is an interesting article that provides local data on the impact of the COVID-19 pandemic on cervical cancer treatment. I have some suggestions that may improve this article. 

Materials and Methods: lines 65-68: Do you know exactly how many patients were excluded because they had only received brachytherapy at CHUM or had only part of their treatment there? It may be helpful to have a figure that displays how many patients were excluded due to various reasons. 

Initially, 291 patients were identified for the study. 47 patients not meeting inclusion criteria were excluded. Figure 1 presenting study design was included. 

lines 80-81: "Date of referral to gynecologic oncology at our center, and date of first consultation by a gynecologic oncologist"; For readability, it may make sense to combine this statement with the previous sentence or edit it to be a complete sentence.  

Agreed, we edited the text. 

lines 82-84: Regarding the date of diagnosis, can you clarify the source for this information? Is it obtained from the referring institution if a patient is referred to CHUM for treatment or further evaluation?  

Clarification made: date of diagnosis (date of first cervical sample confirming cervical cancer) 

line 98: The software reference is for SAS, but I believe IBM still owns SPSS. Please clarify. 

Thank you. The reference was reviewed. 

Lines 134-135: This sentence seems to be a repeat of the sentence at the beginning of the paragraph (at line 130).  

Agreed, we edited the text and removed the repetition “Time between referral and diagnosis did differ pre-post pandemic periods (p=0.042).” 

Table 2: The formatting in this table needs to be fixed so that the data align with the appropriate category labels.  

Thank you. The table formatting was reviewed. 

Discussion: First paragraph: Given this is a single institution study, you may want to frame this paragraph in reference to your experience at a tertiary institution. Additionally, are you able to provide any data that shows your institution is treating most women diagnosed with cervical cancer in a defined geographic area? That may help with generalizability of findings within your geographic region. 

Thank you, edition was made 

lines 221-223: "There was also an important decrease in the number of patients inclusion in the protocol..." I was confused by what you mean here. Is there a way to restate this sentence? "patients included in the protocol?"  

Agreed, edited for “clinical trials” 

Reviewer 3 Report

Comments and Suggestions for Authors

-          I kindly recommend to the Authors to verify the instruction of journal relative to how report the references in the manuscript and tables.

-          The definition of “screening” is unnecessary.

-          Line 53: I suggest changing the term “little” in “scanty/limited”.

-          The aim of the study must be clarified. How the Authors want to measure the “impact of the pandemic”?

-          Which is the period pre and post pandemic? 2018-2020 vs 2020-2022? A sentence should be added.

-          How did they calculate the overall number pre and post pandemic (140 vs 104)? This could be an interesting point in discussion, lead to the difficulties in accessibility to healthcare services. Is it the number of participants in two years? How many invites were made in one year? Which is the percentage of adherence to screening?

-          Did the Authors collect the cytological results or histological results? These results could be reported by cytological classification for pre-cancerous lesions (that are the objectives of the screening program), not only the number of cancers.

-          Lines 144-147: the Authors must specify if the aim is a comparison in management of cancer cases or, differently, they report the impact of pandemic in epidemiology of pre-cancerous or cancerous lesions.

-          The references list must be enlarged in Introduction and Discussion. Considering the moderate impact of pandemic, novel approach should be considered for screening, as self-collection for HPV-testing, with proven accuracy like cervical samples collection. To this, I kindly suggest including a recent research relative to the accuracy of HPV test in vaginal and urine samples: DOI: 10.3390/diagnostics12123075

Comments on the Quality of English Language

Moderate revision

Author Response

-          I kindly recommend to the Authors to verify the instruction of journal relative to how report the references in the manuscript and tables.  

Yes, it has been updated.  

-          The definition of “screening” is unnecessary.  

Agreed, edited 

-          Line 53: I suggest changing the term “little” in “scanty/limited”. 

Agreed, changed for “limited” 

-          The aim of the study must be clarified. How the Authors want to measure the “impact of the pandemic”? 

Agreed, edited: “This is one of the few studies to measure the impact of the pandemic on cervical cancer patients management in a large referral center.” 

-          Which is the period pre and post pandemic? 2018-2020 vs 2020-2022? A sentence should be added.  

The date used for the cutoff is stated at the line 74. All patients from February 2018 until 12 march 2020 were entered the “Before Pandemic” group, while patients from 12 March 2020 until February 2022 entered the “During Pandemic” group.  

-          How did they calculate the overall number pre and post pandemic (140 vs 104)? This could be an interesting point in discussion, lead to the difficulties in accessibility to healthcare services. Is it the number of participants in two years? How many invites were made in one year? Which is the percentage of adherence to screening? 

This study is a single-center retrospective study from February 2018 to February 2022 (line 63). Patients were therefore not invited to participate, instead all patients diagnosed in the study time-period and within the inclusion criteria were reviewed and evaluated by this study. Patients records with exclusion criteria (recurrent diseases, patients who have not received all or part of their treatment following diagnosis at the CHUM, patients who have only received brachytherapy) were not included in the study (line 65-68). To facilitate the comprehension of study design Figure 1 was included.  

-          Did the Authors collect the cytological results or histological results? These results could be reported by cytological classification for pre-cancerous lesions (that are the objectives of the screening program), not only the number of cancers. 

Since we are a reference center and screening was mostly done outside our pathology lab, we found that reporting on the screening results would not provide reliable results. Some patients are referred right after abnormal cytology, while others underwent supplementary screening as colposcopy and others underwent biopsies and LEEPs. The reason for referral can also be biased by the patients’ residency region.  Evaluating Quebec’s screening program was outside the scope of this study.  

-          Lines 144-147: the Authors must specify if the aim is a comparison in management of cancer cases or, differently, they report the impact of pandemic in epidemiology of pre-cancerous or cancerous lesions.  

Edited. The aim is a comparison in management of cancer cases 

-          The references list must be enlarged in Introduction and Discussion. Considering the moderate impact of pandemic, novel approach should be considered for screening, as self-collection for HPV-testing, with proven accuracy like cervical samples collection. To this, I kindly suggest including a recent research relative to the accuracy of HPV test in vaginal and urine samples: DOI: 10.3390/diagnostics12123075 

Text edited and reference added 

Round 2

Reviewer 3 Report

Comments and Suggestions for Authors

We thank the Authors for accepting the suggestions and making the clarifications requested in the text. The manuscript, including the biography and tables, must conform to the journal's standards in order to proceed to publication

Author Response

We have revised the manuscript according to the reviewer's comments.